# Is HOXA5 a Novel Prognostic Biomarker for Uterine Corpus Endometrioid Adenocarcinoma?

**DOI:** 10.3390/ijms241914758

**Published:** 2023-09-29

**Authors:** Changho Song, Kyoung Bo Kim, Gi Su Lee, Soyoung Shin, Byoungje Kim

**Affiliations:** 1Department of Obstetrics and Gynecology, University of Ulsan College of Medicine, Ulsan University Hospital, Ulsan 44033, Republic of Korea; songchanghomd@gmail.com; 2Department of Laboratory Medicine, Keimyung University School of Medicine, Daegu 42601, Republic of Korea; kimbo7072@gmail.com; 3Department of Obstetrics and Gynecology, Keimyung University School of Medicine, Daegu 42601, Republic of Korea; cllgs315@naver.com; 4Department of Pediatrics, Keimyung University School of Medicine, Daegu 42601, Republic of Korea; mizzy94@hanmail.net; 5Department of Radiology, Keimyung University School of Medicine, Daegu 42601, Republic of Korea

**Keywords:** novel prognostic biomarker, uterine corpus, endometrioid adenocarcinoma, endometrial cancer, uterine cancer, HOX genes, gene expression, RNA-sequencing

## Abstract

Endometrial cancer (EC) is one of the most pervasive malignancies in females worldwide. HOXA5 is a member of the homeobox (HOX) family and encodes the HOXA5 protein. HOXA5 is associated with various cancers; however, its association with EC remains unclear. This study aimed to determine the association between HOXA5 gene expression and the prognosis of endometrioid adenocarcinoma, a subtype of EC (EAEC). Microarray data of HOXA5 were collected from the Gene Expression Omnibus datasets, consisting of 79 samples from GSE17025 and 20 samples from GSE29981. RNA-sequencing, clinical, and survival data on EC were obtained from The Cancer Genome Atlas cohort. Survival analysis revealed that HOXA5 overexpression was associated with poor overall survival in patients with EAEC (*p* = 0.044, HR = 1.832, 95% CI = 1.006–3.334). Cox regression analysis revealed that HOXA5 was an independent risk factor for poor prognosis in EAEC. The overexpression of HOXA5 was associated with a higher histological grade of EAEC, and it was also associated with TP53 mutation or the high copy number of EC. Our findings suggest the potential of HOXA5 as a novel biomarker for predicting poor survival outcomes in patients with EAEC.

## 1. Introduction

Endometrial cancer (EC), or uterine cancer, is one of the most prevalent malignancies in females worldwide. In 2020, a total of 417,367 new uterine cancer cases were diagnosed worldwide [1]. The known risk factors for EC include unopposed estrogen exposure, old age, obesity, diabetes mellitus, and atypical endometrial hyperplasia [2,3]. Increases in the use of hormone replacement therapy (HRT), life expectancy, and the prevalence of obesity have led to a global increase in the incidence of EC [4,5]. Fortunately, symptoms of EC, such as vaginal bleeding, tend to present early, resulting in early diagnosis [6,7] and thus early treatment, which greatly improves prognoses [8]. However, even with early diagnosis, certain clinicopathological factors indicate high recurrence rates and poor treatment outcomes [9,10]. The serous adenocarcinoma histologic subtype of EC (SAEC) and high histological grade are common independent risk factors for EC recurrence and are associated with poor survival rates [11,12]. Traditionally, these morphological features have been key factors in assessing the risk of EC recurrence [13]. However, pathologically distinguishing between high-grade EC and SAEC can be challenging [14]. A precise pathological diagnosis is crucial for clinicians when making treatment decisions because early stage high-grade EC may only require adjuvant radiation therapy, whereas SAEC tends to metastasize early and may require systemic chemotherapy even in the early stages [15]. Recent research has enabled the division of EC into four subtypes based on risk stratification according to their molecular features [16]. These four distinct molecular subtypes include DNA polymerase epsilon (POLE) mutations, microsatellite-unstable or mismatch repair deficiency (MSI-h/MMRd), microsatellite-stable (MSS) or low- and high-copy-number ECs, and TP53 mutation or high-copy-number EC [17,18]. Developing these molecular subclassifications improves our understanding of EC diagnostic modalities and creates a potential opportunity for producing targeted therapies [19]. Despite these developments, risk stratification according to molecular variations remains not fully understood.

Homeobox (HOX) genes play substantial roles in cell differentiation and embryogenesis [20,21]. HOXA5 is a member of the HOX family and encodes the HOXA5 protein [22]. HOXA5 inhibits the wingless (Wnt) signaling pathway, and aberrant HOXA5 expression affects tumor cell proliferation, differentiation, invasion, and apoptosis [23,24]. HOXA5 is also associated with various cancers. Hussain et al. reported that HOXA5 expression is elevated in breast cancer [25], and Zhang et al. reported that HOXA5 expression is associated with a poor prognosis in non-small cell lung cancer [26]. Peng et al. reported that HOXA5 is a tumor suppressor gene in gastric cancer [27]. A relationship between HOXA5 and gynecological cancer has also been reported. The downregulation of HOXA5 is associated with poorly differentiated cervical cancer [28,29], and HOXA5 is downregulated in grade 1 EC. However, the association between EC and HOXA5 gene is not yet fully understood [30].

In this study, we investigated the relationship between HOXA5 gene expression and EC prognosis, focusing on the endometrioid adenocarcinoma subtype of EC (EAEC). Additionally, we aimed to integrate HOXA5 gene expression into the traditional classification system based on the molecular subtype of EC. This research may be crucial in contributing to our understanding of the molecular mechanisms underlying endometrial cancer and may pave the way for more accurate prognostic assessments and personalized treatment strategies.

## 2. Results

### 2.1. HOXA5 Is Overexpressed in Endometrial Cancer Tissues

HOXA5 expression in normal and cancerous tissues was analyzed using data from the Gene Expression Omnibus datasets GSE17025 and GSE29981. The GSE17025 dataset included data from 79 patients with stage I EAEC. These diagnoses were substantiated by the Federation of International Gynecology and Obstetrics (FIGO). The 79 samples included 30 grade 1, 33 grade 2, and 16 grade 3 cancer tissues. The dataset GSE29981 included data from 20 healthy endometrial tissue samples. The data for each sample set were merged and normalized before comparing the mean expression levels of the HOXA5 gene. The results showed that the expression level of HOXA5 was significantly higher in cancer tissues than in normal tissues (Figure 1A). Next, we compared the HOXA5 gene expression levels in tissue samples from 23 patients with EC and their paired adjacent normal endometrial tissues. The data were obtained from the TNM plotter online platform (https://TNMplot.com, accessed on 31 August 2023). The expression level of HOXA5 in EC tissue with adjacent normal endometrial tissue was not significantly different (Figure 1B).

### 2.2. HOXA5 Is Associated with Poor Survival in Patients with Endometrial Cancer

The survival data of patients with EC in The Cancer Genome Atlas (TCGA) cohort were subjected to Kaplan–Meier survival analysis using the R survival package. After excluding unavailable gene expression and survival data, 537 patients were included: 196 SAEC and 398 EAEC patients. From the 537 patients with all histological subtypes, we compared 268 patients displaying HOXA5 overexpression and 269 patients demonstrating low expression levels of HOXA5. The results showed that the overexpression of HOXA5 was associated with poor overall survival (OS) in all histological subtypes of EC (*p* = 0.0004, hazard ratio (HR) = 2.159, 95% confidence interval (CI) = 1.390–3.353) (Figure 2). We then compared 199 patients who demonstrated the overexpression of the HOXA5 gene and 199 patients who showed lower HOXA5 gene expression in EAEC, as well as 70 patients who overexpressed the HOXA5 gene and 69 patients who showed a lower expression of the HOXA5 gene in SAEC. According to each histologic subtype, survival analyses showed that HOXA5 was not associated with OS in SAEC (*p* = 0.556, HR = 1.198, 95% CI = 0.657–2.184) but was associated with poor OS in EAEC (*p* = 0.044, HR = 1.832, 95% CI = 1.006–3.334) (Figure 2). The result of the Cox regression analysis of EAEC patients is presented in Table 1. In the univariate analysis, a higher expression of HOXA5, clinical stage, histologic grade and positive cytology were all associated with poorer OS (HOXA5 expression, HR = 2.368, 95% CI = 1.376–4.077, *p* = 0.002; clinical stage, HR = 4.763, 95% CI = 2.862–7.926, *p* < 0.0001; histologic grade, HR = 3.405, 95% CI = 1.809–6.410, *p* < 0.0001; positive cytology, HR = 6.615, 95% CI = 3.739–11.703, *p* < 0.0001, respectively). In the multivariate analysis, higher HOXA5 expression, clinical stage, and positive cytology were identified as independent prognostic factor for poor OS (HOXA5 expression, HR = 2.228, 95% CI = 1.112–4.465, *p* = 0.024; clinical stage, HR = 3.297, 95% CI = 1.652–6.577, *p* = 0.001; positive cytology, HR = 2.667, 95% CI = 1.351–5.265, *p* = 0.005, respectively).

### 2.3. HOXA5 Overexpression Is Associated with a Higher Histological Grade of Endometrial Cancer

After excluding patients with unavailable gene expression data and clinical information, 404 patients with EAEC were included in this study. Age, clinical stage, hypertension, diabetes, menopausal status, history of HRT, postoperative tumor status, postoperative cytology test results, and adjuvant treatment status were not associated with HOXA5 expression. In contrast, histological grade and mean levels of HOXA5 expression were found to be positively correlated (Table 2 and Figure 3). Subgroup analysis was performed by subdividing the three histological grades into group 1 (grade 1), group 2 (grades 2 and 3), group 3 (grades 1 and 2), and group 4 (grade 3). Subgroup analysis revealed that HOXA5 overexpression was associated with a higher histological grade (Table 3). Subsequently, we performed receiver operating characteristic (ROC) curve analysis with 213 grade 1 and 2 EAEC patients, and 189 grade 3 EAEC patients. The result of ROC curve analysis using data from 213 grade 1 and 2 EAEC patients and 189 grade 3 EAEC patients indicated that HOXA5 could be utilized to discriminate high-grade EC, although it was not considered an ideal tool for distinguishing high-grade EC (AUC = 0.644. 95% CI, 0.598–0.690). The ROC curve analysis suggested an optimal cut-off value of 2.019, with a sensitivity of 0.586 and specificity of 0.636 (Figure 4).

Our findings suggest that there is indeed an association between HOXA5 gene expression and EC prognosis. Moreover, HOXA5 can be used for discriminating high-grade EC. The overexpression of HOXA5 is associated with a higher histological grade; this is one of the key risk factors for EC recurrence and may lead to poor OS in patients with EC.

### 2.4. HOXA5 Is Overexpressed in the High-Copy-Number Endometrial Carcinoma Group

By incorporating novel molecular classification, patients were sub-grouped into four categories: POLE mutation, MSI-h/MMRd, MSS or normal-copy-number, and TP53 mutation or high-copy-number groups. The Kaplan–Meier survival analysis showed the most favorable OS in the POLE mutation group and the worst OS in the TP53 mutation or high-copy-number group (Figure 5A). HOXA5 expression was the highest in the TP53 mutation or high-copy-number group (Figure 5B).

## 3. Discussion

In this study, we determined the prognostic value of HOXA5 gene expression in EC. EC is the most common gynecological malignancy in developed countries, and its incidence is increasing worldwide [31]. The symptoms of EC tend to present early, leading to early diagnosis; however, EC still causes approximately 90,000 cancer-related deaths annually [32]. Age, race, obesity, nulliparity, diabetes, hypertension, HRT, histologic subtype, and histologic grade are the known prognostic factors for EC [33].

In 2013, TCGA (http://www.cancer.gov/about-nci/organization/ccg/research/structure-genomics/tcga/using-tcga/citing-tcga, accessed on 1 January 2023), which is a large-scale genomic analysis of various types of cancer, enabled further understanding of the molecular and genomic aspects of cancer [34]. Through this research, crucial information, such as whole-exome sequencing, somatic copy number alterations, methylation profiles, and the somatic mutations of patients with EC, became available to the public. As a result, researchers have identified four distinct molecular features of EC: POLE mutations, microsatellite instability or mismatch repair deficiency, and low- and high-copy-number ECs [35]. However, further clarification is needed to better understand how these molecular variations relate to traditional classification and the assessment of risk factors [16].

The HOX genes are found in almost all eukaryotic cells and were first identified in 1992 [36,37]. The genes typically consist of a highly conserved DNA sequence of 180 base pairs and encode the homeodomain, which is a protein domain that binds to specific DNA sequences [38]. The HOX gene family plays a key role in cell differentiation and embryogenesis [20,21] and is involved in the development and healthy functioning of the female reproductive tract [39]. Studies have identified that the dysregulation of HOX genes is associated with many types of cancers [36,40]. The aberrant expression of HOX genes may affect apoptosis, angiogenesis, receptor signaling, and differentiation, resulting in the promotion of oncogenesis or tumor suppression [41]. In humans, the HOX gene cluster can be divided into four groups: HOXA, HOXB, HOXC, and HOXD [42]. Each group contains a series of HOX genes, and HOXA5 is part of the series within the HOXA cluster. Other series within the HOXA cluster have been identified as oncogenes or tumor suppressors in various cancers [41]. HOXA1 is known to be an oncogene in breast cancer, glioma, and gastric cancer [43,44,45]. HOXA3 is known to be an oncogene in non-small cell lung cancer and thyroid cancer [46,47]. HOXA6 and HOXA13 promote gastric cancer and colorectal cancer [48,49]. HOXA7 has been found to be associated with the development and progression of cervical cancer and hepatocellular carcinoma [50,51]. HOXA9 induces breast cancer and leukemia, and HOXA10 is an oncogene of prostate and testicular cancers [52,53,54]. Finally, HOXA11 is known to promote gastric cancer and renal cancer [55]. In contrast, HOXA4 is known to be a tumor suppressor gene in lung and ovarian cancers [56,57]. These studies indicate that HOXA genes can act as oncogenes or tumor suppressors; therefore, HOXA genes might be novel therapeutic targets for the treatment and prevention of cancer. Importantly, however, the prognostic value and clinical significance of HOXA expression in EC remain unclear.

After comparing the expression levels of HOXA5 in normal and cancerous tissues using the GSE17025 and GSE29981 datasets, we observed that HOXA5 was overexpressed in EC tissues compared to normal endometrial tissues. However, the HOXA5 expression level did not show difference when comparing EC tissue and paired normal endometrial tissue. The inconsistency in these results may be attributed to the relatively small sample size or the presence of tissue heterogeneity. A larger-scale study is need in the future to validate these results. Survival analyses showed that HOXA5 overexpression was associated with poor survival in patients with EC for all histological subtypes. Specifically, HOXA5 overexpression was associated with poor survival in patients with EAEC but not in patients with SAEC. The mean expression level of the HOXA5 gene was positively associated with the histological grade of EAEC. Moreover, Cox regression analysis demonstrated that a higher HOXA5 expression, clinical stage, and positive cytology were independent risk factors for poor OS. ROC curve analysis showed that HOXA5 could discriminate high-grade EC but with limited accuracy.

In our study, we analyzed HOXA5 expression according to molecular classification, the POLE gene functions in DNA duplication, and tumor suppression [58]. There were 80 patients who showed POLE gene mutations, with the most common mutations being Val411Leu (n = 13) and Pro286Arg (n = 21). Although grade 3 EC (n = 51) was the most common in the POLE mutation group, it exhibited the most favorable OS and a relatively low expression of HOXA5 compared to the other groups.

Microsatellites are short, repeated DNA sequences, and defects in MMR function can lead to microsatellite instability [59]. MSI-h/MMRd is often determined via the immunohistochemical staining of MMR proteins, such as MLH1, MSH2, MSH6, and PMS2. It can also be detected by the hypermethylation of the MLH1 promotor area [60,61,62]. In our study, there were 90 patients in the MSI-h/MMRd group, most of whom had grade 1 or 2 EAEC. Next, patients with confirmed TP53 gene mutations or high somatic copy numbers were grouped separately. The TP53 gene mutation or high-copy-number group consisted of 78 and 40 patients with SAEC and EAEC, respectively. Among the 40 patients with EAEC, 32 had high-grade EC. This group showed the worst prognosis in the survival analysis.

There were also 251 patients who had MSS or normal copy numbers. The survival analysis yielded similar results to a study by the Cancer Genome Atlas Research Network et al. (2013), although the proportion of the POLE mutation group was slightly higher in our study [35]. The POLE mutation group showed the most favorable survival outcome, whereas the TP53 mutation or high-copy-number group showed the worst survival outcome. The expression of HOXA5 did not show a precise correlation with this molecular subtyping, but we observed the overexpression of HOXA5 in the TP53 gene mutation or high-copy-number group, which was considerably associated with poor survival outcomes. It is important to note that owing to limitations in interpreting publicly available data, our molecular subgrouping method may not have been identical to previous reports [35]. Nevertheless, the results of our study suggest that HOXA5 overexpression is a potential biomarker, indicating a poor prognostic outcome in patients with EC. Moreover, we have shed light on a novel biomarker for predicting the prognosis of patients with EC that incorporates both traditional risk factors and new molecular classification.

Aberrant HOXA5 expression has been associated with various cancers [23]. HOXA5 is downregulated in breast cancer, gastric cancer, colorectal cancer, hepatocellular carcinoma, lung cancer, osteosarcoma, and adrenocortical carcinoma but is overexpressed in oral squamous carcinoma, esophageal squamous carcinoma, glioma, and leukemia [23]. Some studies have shown a relationship between aberrant HOXA5 gene expression and gynecologic cancers, such as cervical cancer and EC. In these reports, HOXA5 overexpression was associated with a better prognosis in cervical cancer [28,29]. One study reported that HOXA5 was downregulated in the glandular tissue of grade 1 EC [30]. Conversely, our study suggests that the overexpression of HOXA5 is associated with higher histological grade, TP53 mutation or high-copy-number EC, and poor survival in patients with EAEC.

Our study’s strength lies in its comprehensive analysis of the microarray and RNA-sequencing data of patients accumulated from different databases. Moreover, we analyzed HOXA5 expression levels according to novel molecular classification. To the best of our knowledge, this is the first report to indicate that HOXA5 overexpression is associated with poor survival in patients with EC. One of the limiting factors of this study is its retrospective analysis of the published data. To thoroughly explore the use of HOXA5 as a therapeutic target for patients with EC, further studies are necessary to validate these results and reveal the molecular pathways of HOXA5 in EC pathophysiology.

## 4. Materials and Methods

### 4.1. Acquisition of Microarray Datasets

The gene expression microarray datasets GSE17025 and GSE29981 were downloaded from the publicly available Gene Expression Omnibus database (National Institutes of Health, Bethesda, MD, USA; http://www.ncbi.nlm.nih.gov/geo, accessed on 3 February 2023). The GSE17025 dataset included data from 79 tissues from patients with stage I EAEC, whereas the GSE29981 dataset included data from 20 healthy endometrial tissues. Both samples were analyzed using an Affymetrix GeneChip Human Genome U133 plus 2.0 Array (Affymetrix, Santa Clara, CA, USA). The Affymetrix ID 213844_at (HOXA5) was valid. The basic dataset information is presented in Table 4.

### 4.2. Data Normalization

A robust multiarray average algorithm and a quantile normalization method were used to normalize the data. Differences in gene expression between normal and cancer tissues were analyzed using Student’s *t*-test. The box plots displaying the differential gene expression levels between normal and cancer tissues, as well as the log2-fold change, were plotted using the R programming language (version 3.4.1; http://cran.r-project.org/, accessed on 7 March 2023). Statistical significance was set at *p* < 0.05.

### 4.3. Acquisition and Analysis of Clinical Data

The RNA-sequencing datasets were downloaded from the USCS Xena Browser (http://xenabrowser.net/, 1 January 2023) and included gene expression (dataset ID: TCGA-UCEC.Htseq_fpkm), clinicopathological parameters (dataset ID: TCGA-UCEC.GDC_phenotype), and survival data (dataset ID: TCGA-UCEC.survival) from patients with EC. Of the 583 RNA-sequencing data samples, 139 SAEC and 398 EAEC samples were included in the survival analysis after excluding those with insufficient survival data. A total of 402 patients with EAEC were analyzed after excluding samples with insufficient clinical data. The clinical data included age, clinical stage, hypertension, diabetes, menopausal status, history of HRT, postoperative tumor status, postoperative cytology test results, adjuvant treatment status, and survival information. The mean expression of HOXA5 was analyzed according to each clinical parameter. For survival analysis, the patients were divided into high- and low-gene expression groups according to the median gene expression level. Survival analysis was performed using the Kaplan–Meier survival and Cox regression analyses using the survival package (version 3.5-5; http://CRAN.Rproject.org/package=survival, accessed on 27 April 2023) in R (version 4.3.0; http://cran.r-project.org/, accessed on 27 April 2023). This study met the publication guidelines provided by TCGA. The data for the comparison of gene expression between EC tissues and paired adjacent normal tissues were downloaded from the TNM plotter database. TNM plotter is an online platform that provides a comparison of gene expression levels between tumor or metastatic tissues and paired or non-paired normal tissues [65]. The RNA-sequencing data of the HOXA5 gene from 23 patients with EC and their paired adjacent normal tissues were downloaded and analyzed.

### 4.4. Subgrouping According to Molecular Classification

The molecular classification datasets were downloaded from the USCS Xena Browser and included somatic mutation (dataset ID: TCGA-UCEC.muse_snv), methylation data (dataset ID: TCGA-UCEC.methylation450), and copy number data (dataset ID: TCGA-UCEC.cnv). First, we grouped the patients based on POLE gene mutations. The mutations considered for this grouping included intron variants, missense variants, synonymous variants, splice acceptor variants, and splice region variants of the POLE gene. Next, we grouped patients based on the hypermethylation of the MLH1 promotor region, which was defined as a high methylation beta value of 0.9512 when using the Illumina Infinium HumanMehtlyation450 Beadchip. Finally, we grouped patients based on either TP53 gene mutations or the high-level amplification of TP53 gene copies. Patients with normal copy numbers or patients without the hypermethylation of the MLH1 promotor region were grouped separately.

### 4.5. Statistical Analyses

The R programming language was used to analyze the data. One-way analysis of variance (ANOVA) was used to compare the mean expression of genes among three or more groups. Post hoc tests were conducted using Bonferroni correction when significant results were observed. The results were considered statistically significant at *p* < 0.05. Gene expression levels in normal and tumor tissues were compared using Levene’s test and the Student’s *t*-test. A *p*-value < 0.05 in the Levene’s test was considered to indicate a non-parametric distribution of variances. A *p*-value < 0.05 was considered statistically significant when using the Student’s *t*-test. For the survival analysis, the patients were divided into higher expression and lower expression by using the median HOXA5 expression level as a cutoff point. The Kaplan–Meier survival analysis was used to analyze the data. The Cox proportional hazard model was utilized in the univariate and multivariate analysis. Variables that exhibited a *p*-value less than 0.05 in the univariate analysis were subsequently included in the multivariable analysis. ROC curve analysis was conducted using SPSS (IBM SPSS Statistics for Windows, version 26.0; IBM Corp, Armonk, NY, USA). The results were considered statistically significant at *p* < 0.05, and an area under the curve greater than 0.7 was considered indicative of a good prognostic model.

## 5. Conclusions

Our study identified that HOXA5 was associated with a higher histological grade and poor survival in patients with EAEC. These findings may provide new insights into the pathophysiology of EAEC and may have broad implications in developing future clinical prognostic tools. Further investigations are needed to validate these outcomes, but we carefully propose the potential utilization of HOXA5 as a novel biomarker for predicting poor survival outcomes in patients with EAEC.

## Figures and Tables

**Figure 1 ijms-24-14758-f001:**
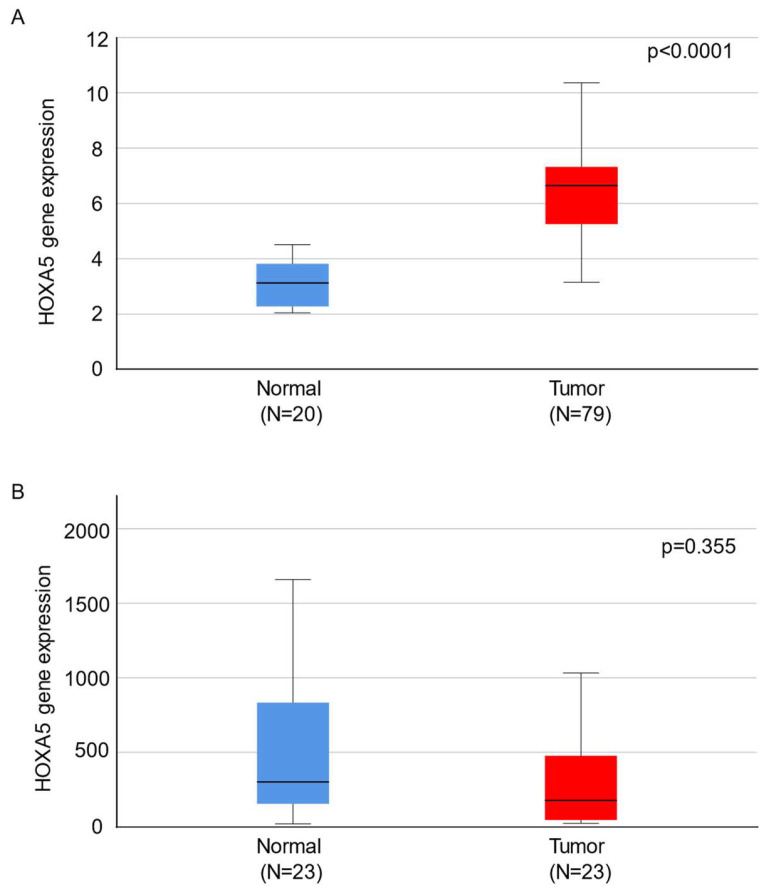
Box plots showing the relative expression of HOXA5 between normal and cancerous tissues. (**A**) Comparison of the microarray data of 79 endometrial cancer tissues and 20 non-adjacent normal endometrial tissues. (**B**) Comparison of the RNA-sequencing data of 23 endometrial cancer tissues and 23 paired adjacent normal endometrial tissues.

**Figure 2 ijms-24-14758-f002:**
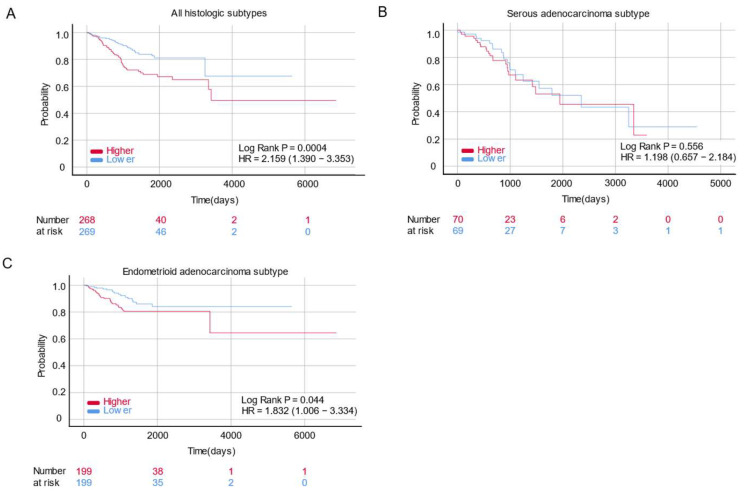
Survival analysis of HOXA5 gene expression in patients with endometrial cancer. (**A**) Kaplan–Meier curve of patients with endometrial cancer. All histological subtypes were included (endometrioid and serous adenocarcinomas) according to the relative mRNA expression levels of the HOXA5 gene. (**B**) Kaplan–Meier curve of patients with endometrial cancer. The histological subtype serous adenocarcinoma is displayed according to the relative mRNA expression levels of the HOXA5 gene. (**C**) Kaplan–Meier curve of patients with endometrial cancer. The histological subtype endometrioid adenocarcinoma is displayed according to the relative mRNA expression levels of the HOXA5 gene.

**Figure 3 ijms-24-14758-f003:**
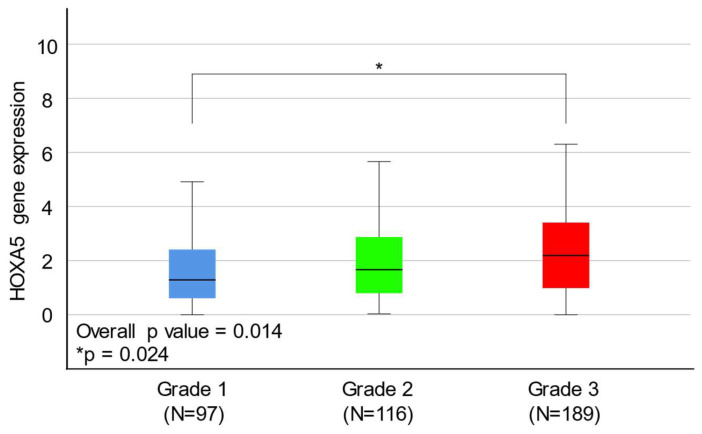
Box plot showing the relative expression levels of the HOXA5 gene according to histological grade in the endometrioid adenocarcinoma subtype acquired from patients with endometrial cancer.

**Figure 4 ijms-24-14758-f004:**
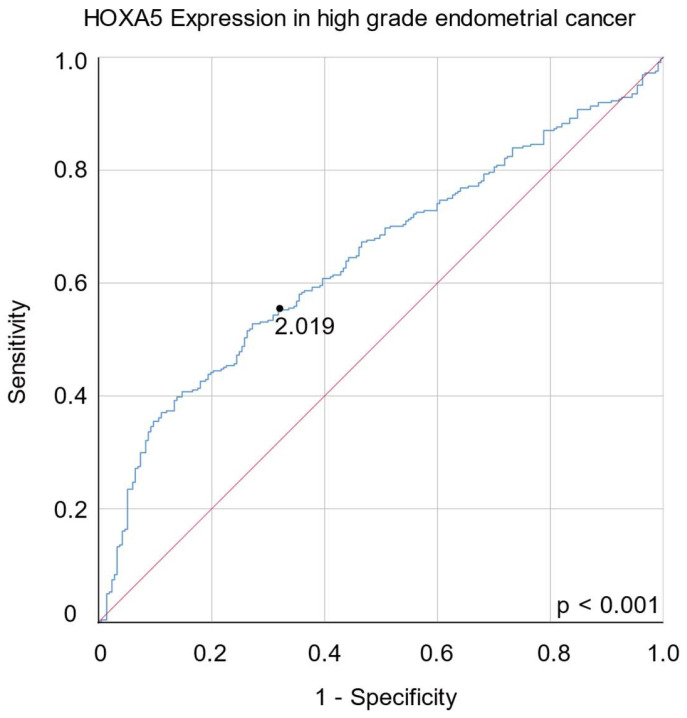
Receiver operating characteristic curve analysis of HOXA5 gene expression for discrimination between high-grade and low-grade endometrial cancers.

**Figure 5 ijms-24-14758-f005:**
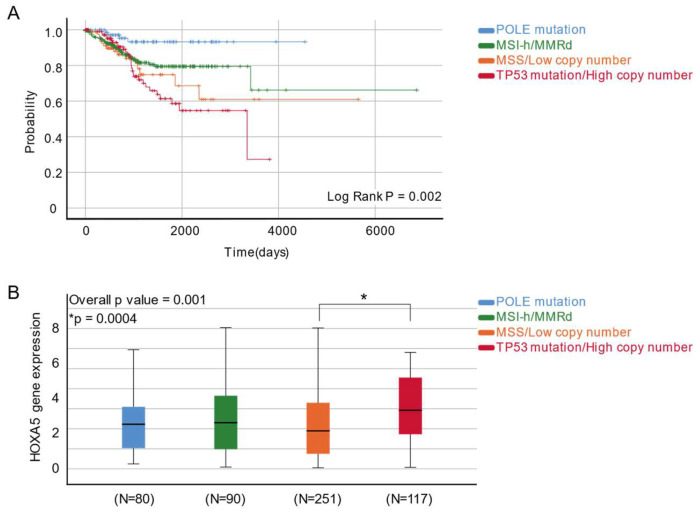
Survival analysis and HOXA5 expression levels in patients with endometrial cancer according to molecular classification. (**A**) Kaplan–Meier curve of patients with endometrial cancer, with subgrouping by POLE mutation, MSI-h/MMRd, MSS/normal copy number, and TP53 mutation/high copy number. (**B**) Comparison of mean HOXA5 gene expression levels, with subgrouping by POLE mutation, MSI-h/MMRd, MSS/normal copy number, and TP53 mutation/high copy number.

**Table 1 ijms-24-14758-t001:** Cox regression analysis of overall survival.

Variables		Univariate			Multivariate		
		HR ^†^	95% CI ^‡^	*p*-Value	HR ^†^	95% CI ^‡^	*p*-Value
HOXA5	Lower	1	-	-	-	-	-
expression	Higher	2.368	1.376–4.077	0.002	2.286	1.129–4.630	0.022
Age	<60	1	-	-	-	-	-
	≥60	1.678	0.923–3.049	0.09	0.872	0.436–1.746	0.699
Stage	1, 2	1	-	-	-	-	-
	3, 4	4.763	2.862–7.926	<0.0001	3.275	1.640–6.541	0.001
Diabetes	No	1	-	-	-	-	-
	Yes	0.959	0.472–1.947	0.908	-	-	-
Hypertension	No	1	-	-	-	-	-
	Yes	0.924	0.508–1.680	0.796	-	-	-
HRT ^§^	No	1	-	-	-	-	-
	Yes	0.891	0.339–2.344	0.815	-	-	-
Menopause	No	1	-	-	-	-	-
	Yes	1.202	0.481–3.004	0.693	-	-	-
Grade	1, 2	1	-	-	-	-	-
	3	3.405	1.809–6.410	<0.0001	1.898	0.887–4.062	0.099
Cytology	Negative	1	-	-	-	-	-
	Positive	6.615	3.739–11.703	<0.0001	2.734	1.367–5.470	0.004

^†^ Hazard ratio; ^‡^ confidential interval; ^§^ hormone replacement therapy.

**Table 2 ijms-24-14758-t002:** The single variable analysis of mean HOXA5 expression according to clinical parameters.

Parameters		Number	Mean Gene Expression (FPKM ^†^)	*p*-Value
Age	<60	160	1.812	0.273
	≥60	239	1.980	
Clinical stage	Stage I and II	286	1.909	0.792
	Stage III and IV	116	1.953	
Histological grade	Grade 1	97	1.655	0.014
	Grade 2	116	1.773	
	Grade 3	189	2.150	
Hypertension	No	120	1.818	0.995
	Yes	174	1.819	
Diabetes	No	189	1.877	0.335
	Yes	79	1.700	
HRT ^‡^	No	184	1.679	0.154
	Yes	28	2.047	
Menopausal status	Pre	36	2.100	0.430
	Peri	32	1.631	
	Post	313	1.919	
Cytology	negative	271	1.937	0.807
	positive	28	2.011	
Tumor status	Negative tumor	310	1.797	0.055
	With tumor	42	2.250	
Adjuvant treatment	No	289	1.850	0.721
	Yes	94	1.912	

^†^ Fragment per kilobase million; ^‡^ Hormone replacement therapy.

**Table 3 ijms-24-14758-t003:** Subgroup analysis of HOXA5 gene expression and histological grade.

Subgroup		Number	Mean Gene Expression (FPKM ^†^)	*p*-Value
Group 1 vs. 2	Group 1 (grade 1)	97	1.655	0.028
	Group 2 (grades 2 and 3)	305	2.007	
Group 3 vs. 4	Group 3 (grades 1 and 2)	213	1.369	0.004
	Group 4 (grade 3)	289	1.613	

^†^ Fragment per kilobase million.

**Table 4 ijms-24-14758-t004:** Microarray dataset information from the NCBI Gene Expression Omnibus (GEO) database.

Platform	GEO Dataset	Samples	Reference
GPL570	GSE17025	79 EAEC ^†^	Day, R. S. et al. [63,64]
	GSE29981	20 healthy EM ^‡^	

^†^ Endometrioid adenocarcinoma subtype of endometrial cancer; ^‡^ endometrium.

## Data Availability

The data of this study are available from the corresponding author upon request.

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
