# Peer review of "Is HOXA5 a Novel Prognostic Biomarker for Uterine Corpus Endometrioid Adenocarcinoma?"

_ijms, 2023, doi:10.3390/ijms241914758_

Round 1

Reviewer 1 Report

Title: HOXA5, a novel prognostic biomarker for uterine corpus endometrioid adenocarcinoma

Authors: Song et al.

General comments

The current is an experimental study aimed to assess the role played by HOXA5 in endometrial cancer with regard for its prognosis. HOXA5 resulted overexpressed in endometrioid adenocarcinoma of endometrium, especially in cancer with high histological grade, and resulted associated to poor survival outcomes. The study is interesting, but it needs careful revision.

Specific comments

Please specify in the title and in the text that your data regard stage I endometrial cancer patients. This is particularly true for the conclusion.

The abstract section should include quantitative data including the sample size and the correlation data with survival.

Data on normal endometrial tissue are totally lacking in the methods. Please detail your inclusion/exclusion criteria

It should be very interesting to study the HOXA5 expression according to new molecular subclassifications, as detailed in the Introduction. Please detail these data for your 79 cases.

A ROC for HOXA5 gene expression could be useful. I suggest building your ROC and to perform an attempt as prognostic tool in your population and setting. The efficacy of the use of combined risk factors could be also interesting.

Author Response

Title: HOXA5, a novel prognostic biomarker for uterine corpus endometrioid adenocarcinoma

Authors: Song et al.

General comments

The current is an experimental study aimed to assess the role played by HOXA5 in endometrial cancer with regard for its prognosis. HOXA5 resulted overexpressed in endometrioid adenocarcinoma of endometrium, especially in cancer with high histological grade, and resulted associated to poor survival outcomes. The study is interesting, but it needs careful revision.

Specific comments

Please specify in the title and in the text that your data regard stage I endometrial cancer patients. This is particularly true for the conclusion.

Author’s response: Thank you for your valuable comment. As you advised, we have modified the title and in the manuscript.

The abstract section should include quantitative data including the sample size and the correlation data with survival.

Author’s response: Thank you for your valuable comment. As you advised, we have added quantitative data and correlation data with survival in the abstract section.

Data on normal endometrial tissue are totally lacking in the methods. Please detail your inclusion/exclusion criteria

Author’s response: Thank you for your kind comment. The design of this study was analyzing difference among endometrial cancer patients according to histologic grade, histologic subtype, and clinical stage etc. We analyzed data of cancer patients downloaded from TCGA cohort. The data of normal endometrial tissue was analyzed and presented in comparison to endometrial cancer tissue by downloading data from GEO database which used microarray method and downloading RNA sequencing data from TNM plotter which is an online platform.

It should be very interesting to study the HOXA5 expression according to new molecular subclassifications, as detailed in the Introduction. Please detail these data for your 79 cases.

Author’s response: We appreciate your comment. This is very valuable comment. We analyzed HOXA5 expression data in accordance with novel molecular subclassification. We have identified overexpression of HOXA5 in TP53 mutation or high copy number group. The results are added in the manuscript. Once again, thank you for your comment. 

A ROC for HOXA5 gene expression could be useful. I suggest building your ROC and to perform an attempt as prognostic tool in your population and setting. The efficacy of the use of combined risk factors could be also interesting.

Author’s response: We appreciate your comment. As you advised, we have performed ROC analysis and the result was added in the manuscript.

Reviewer 2 Report

The manuscript presented by Kim and coworkers explores the role of HOXA5 in endometrioid adenocarcinoma using published datasets of microarray and RNA-Seq. The main criticism of the study is the lack of data validation by analyzing HOXA5 in an independent cohort of samples. A proper validation should be made by the analysis of a set of samples (the best choice) or by analyzing an independent data set. 

Minor concerns:

1. In addition to Kaplan-Meier survival analysis I suggest performing the multivariate Cox's regression including in addition to HOXA5 expression also clinical and pathological data of the patients in order to estimate the influence of covariates to the survival.

2. Table 1: Please, specify if results refer to multivariable analysis or single variable analysis.

3. Figure 3 should be modified using only panel A where grade 1, 2 and 3 are reported. The other grouping system is not relevant.

There are only some minor typing errors.

Author Response

The manuscript presented by Kim and coworkers explores the role of HOXA5 in endometrioid adenocarcinoma using published datasets of microarray and RNA-Seq. The main criticism of the study is the lack of data validation by analyzing HOXA5 in an independent cohort of samples. A proper validation should be made by the analysis of a set of samples (the best choice) or by analyzing an independent data set. 

Minor concerns:

  1. In addition to Kaplan-Meier survival analysis I suggest performing the multivariate Cox's regression including in addition to HOXA5 expression also clinical and pathological data of the patients in order to estimate the influence of covariates to the survival.

Author’s response: We are thankful to your comment. As you advised, we have performed univariate and multivariate Cox regression. The result is in new table 1. Once again, thank you for your valuable comment.

  1. Table 1: Please, specify if results refer to multivariable analysis or single variable analysis.

Author’s response: We appreciate your comment. As you advised, we have specified that table 1 is single variable analysis. 

  1. Figure 3 should be modified using only panel A where grade 1, 2 and 3 are reported. The other grouping system is not relevant.

Author’s response: We appreciate your comment. As you advised, we have removed Figure 3b and Figure 3c.

Comments on the Quality of English Language

There are only some minor typing errors. 

Authors response: Thank you, we have gone through a new English editing.

Reviewer 3 Report

In this research article, Changho Song et al. mainly used databases to analyze the correlation of HOXA5 and endometrial cancer progression and prognosis. This paper cannot be accepted without the following modifications:

1.      There was too much irrelevant information included in the Introduction, like the EC early diagnosis, stratification, etc. have nothing to do with the article, should be simplified and merged.

2.      Why mainly focused on endometrioid adenocarcinoma subtype of EC (EAEC)? Any background information about this subtype?

3.      All the figures are of low resolution, which makes them hard to read.

4.      In Fig1, the comparison of HOXA5 expression should be compared between the cancerous area and the paired untransformed healthy tissue, the data the authors used in Fig1 was collected from 2 independent databases, which were not comparable.

5.      There was SAEC subtype mentioned in Fig2 even though the authors illustrated in the Introduction that this article would be mainly focusing on EAEC.

6.      In Fig2, what’s the definition of ‘overexpression’ and ‘low expression’? Do any IHC and histological scores confirm that?

7.      There were no actual experiments done for this paper whatsoever. All the data was acquired from GSE, TCGA etc. There wasn’t sufficient amount of convincing data to draw any solid conclusions, and the quality of current data should be drastically improved.

Minor editing of English language required.

Author Response

  1. There was too much irrelevant information included in the Introduction, like the EC early diagnosis, stratification, etc. have nothing to do with the article, should be simplified and merged.

Author’s response: We appreciate your comment. As you advised, we have modified our article with removing irrelevant information and rewriting the manuscript.

  1. Why mainly focused on endometrioid adenocarcinoma subtype of EC (EAEC)? Any background information about this subtype?

Author’s response: Thank you for your valuable comment. SAEC accounts up to 10% of EC but SAEC itself is a risk factor of poor prognosis. However, EAEC is most common type of EC and vast majority of it is low grade EC and presents in an early stage. Identifying the risk factors of EAEC is important for establishing adjuvant treatment plan. As you advised, we’ve added mentioned above in the manuscript to clarify the reason why we are focusing on EAEC. Thank you.   

  1. All the figures are of low resolution, which makes them hard to read.

Author’s response: We appreciate your comment. Figures of this article was constructed using vector-based graphic tool. There must have been technical error. We will provide high resolution image. We apology for your inconvenience.

  1. In Fig1, the comparison of HOXA5 expression should be compared between the cancerous area and the paired untransformed healthy tissue, the data the authors used in Fig1 was collected from 2 independent databases, which were not comparable.

Author’s response: We appreciate your comment. Like you mentioned, it is best to compare data from same experiment and using cancer tissue to paired normal tissue. Using paired tissue is useful for identifying genes particularly dysregulated in cancer. But using cancer tissue and unrelated individuals’ normal tissue can have advantages in identifying cancer related gene expression changes on a broader scale across different populations. In our study, we analyzed data produced by other experiments. We tried to minimize bias as possible. We used Micro-array data of normal and cancer tissue using same platform GPL570 (Affymetrix Genechip Human Genome U133 plus 2.0 Array). All expression profiling data were merged, and background correction followed by normalization was conducted using Robust Multi-array Average (RMA) algorithm and quantile normalization. We think it is possible to compare data from different experiment by going through a thorough data preprocessing, normalization and using proper statistical method. By taking your advice, we added data comparing HOXA5 expression data from EM cancer tissue and paired normal endometrial tissue in figure 1.

  1. There was SAEC subtype mentioned in Fig2 even though the authors illustrated in the Introduction that this article would be mainly focusing on EAEC.

Author’s response: We appreciate your comment. The reason we presented the Kaplan-Meier survival curve of SAEC in figure 2 was to emphasize that HOXA5 is particularly useful in predicting prognosis of EAEC. We agree with you, this can be removed if it seems unnecessary. 

  1. In Fig2, what’s the definition of ‘overexpression’ and ‘low expression’? Do any IHC and histological scores confirm that?

Author’s response: We are very thankful to your kind comment. The patients were grouped into higher and lower expression groups by dividing them at a cutoff value of the median expression of HOXA5 gene. To clarify the meaning, we have added mentioned above in method section.

  1. There were no actual experiments done for this paper whatsoever. All the data was acquired from GSE, TCGA etc. There wasn’t sufficient amount of convincing data to draw any solid conclusions, and the quality of current data should be drastically improved.

Author’s response: We are very thankful to your kind comment. As you mentioned there are limitations of our study. We have clearly stated our limitation in the manuscript and modify the conclusion to conclusion with lower-level evidence.  

Comments on the Quality of English Language

Minor editing of English language required.

Authors response: Thank you, we have gone through a new English editing.

Reviewer 4 Report

I read with great interest the Manuscript titled " HOXA5, a novel prognostic biomarker for uterine corpus endometrioid adenocarcinoma”, topic interesting enough to attract readers' attention.

Although the manuscript can be considered already of good quality, I would suggest following recommendations: 

-    I suggest a round of language revision, in order to correct few typos and improve readability

-       In recent years the classification of endometrial cancer has evolved significantlyIt would be interesting to discuss results of this study in the contest of current evidence about molecular insights of endometrial cancer, considering prognosis and the possibility of tailored management. I would be glad if the authors discuss this important point, referring to PMID: 36833105 and 36979434.

Because of these reasons, the article should be revised and completed. Considering all these points, I think it could be of interest to the readers and, in my opinion, it deserves the priority to be published after minor revisions.

I suggest a round of language revision, in order to correct few typos and improve readability

Author Response

I read with great interest the Manuscript titled " HOXA5, a novel prognostic biomarker for uterine corpus endometrioid adenocarcinoma”, topic interesting enough to attract readers' attention.

Although the manuscript can be considered already of good quality, I would suggest following recommendations: 

-    I suggest a round of language revision, in order to correct few typos and improve readability

-       In recent years the classification of endometrial cancer has evolved significantly. It would be interesting to discuss results of this study in the contest of current evidence about molecular insights of endometrial cancer, considering prognosis and the possibility of tailored management. I would be glad if the authors discuss this important point, referring to PMID: 36833105 and 36979434.

Because of these reasons, the article should be revised and completed. Considering all these points, I think it could be of interest to the readers and, in my opinion, it deserves the priority to be published after minor revisions.

Comments on the Quality of English Language

I suggest a round of language revision, in order to correct few typos and improve readability

Author’s response: We are very thankful to your kind comment. We have gone through a new English editing. Like you advised, we analyzed HOXA5 expression data in accordance with novel molecular subclassification. We have identified that there was overexpression of HOXA5 in TP53 mutation or high copy number group. The results are added in the manuscript. Once again, we appreciate your valuable comment. And as you advised, we went through an another round of English editing.   

Round 2

Reviewer 1 Report

The suggestions/comments have been correctly followed.

The manuscript has been sufficiently improved.

Author Response

We appreciate for your kind comments.

Reviewer 2 Report

The authors have tried to answer the question I posed. Nevertheless, I have some minor concerns:

1. Table 1: In the Cox regression model (multivariate), all the covariates presented in the table should be included, not only grade, HOXA5, and cytology. Please repeat the analysis and fill the gaps in Table 1.

2. Can the authors comment on the findings from the TNM plot analysis? 

There are some sentences to rephrase or correct:

Rows 141-143: please rephrase.

Rows 229-232, please correct as follows: Moreover, Cox regression analysis demonstrated that higher HOXA5 expression, the clinical stage, and positive cytology were independent risk factors for poor OS. ROC curve analysis showed that HOXA5 discriminated high-grade EC, but with limited accuracy.

Author Response

The authors have tried to answer the question I posed. Nevertheless, I have some minor concerns:

  1. Table 1: In the Cox regression model (multivariate), all the covariates presented in the table should be included, not only grade, HOXA5, and cytology. Please repeat the analysis and fill the gaps in Table 1.

Author’s response: We appreciate your comment. We have filled the gaps in Table 1 to the best of our ability. However, we were unable to provide data for variables such as menopause, diabetes, hypertension, and hormone replacement therapy (HRT) since these variables were not included in the multivariate analysis. We believe that multivariate cox regression analysis is performed to analyze independent effects of each variable on survival outcome while adjusting for the influence of other variables. Including statistically significant variables from univariate analysis is a common practice, and it can help ensure that the variables included in the multivariate model have a meaningful impact on the outcome and are less likely to introduce noise into the analysis. However, even if variables from univariate analysis did not show statistical significance, they can still be included in the multivariate analysis if there is a theoretical reason or prior knowledge suggesting their clinical or biological importance.

In response to your advice, we filled the gap with a variable that showed insignificance in the multivariate analysis. Additionally, to address the missing data in Table 1, we added age as a variable, which, although statistically insignificant, is known as a major risk factor in endometrioid adenocarcinoma of the endometrium based on the multivariate analysis. We hope this helps provide a more comprehensive overview of the study's findings.

  1. Can the authors comment on the findings from the TNM plot analysis? 

Author’s response: We appreciate your comment. As you advised, we have added comment in discussion section.

Comments on the Quality of English Language

There are some sentences to rephrase or correct:

Rows 141-143: please rephrase.

Author’s response: We appreciate your comment. As you advised, we have rephrased the sentence.

Rows 229-232, please correct as follows: Moreover, Cox regression analysis demonstrated that higher HOXA5 expression, the clinical stage, and positive cytology were independent risk factors for poor OS. ROC curve analysis showed that HOXA5 discriminated high-grade EC, but with limited accuracy.

Author’s response: We appreciate your comment. We have rephrased the sentence as advised.
